# Coumarinolignans with Reactive Oxygen Species (ROS) and NF-κB Inhibitory Activities from the Roots of *Waltheria indica*

**DOI:** 10.3390/molecules27103270

**Published:** 2022-05-19

**Authors:** Feifei Liu, Sudipta Mallick, Timothy J. O’Donnell, Ruxianguli Rouzimaimaiti, Yuheng Luo, Rui Sun, Marisa Wall, Supakit Wongwiwatthananukit, Abhijit Date, Dane Kaohelani Silva, Philip G. Williams, Leng Chee Chang

**Affiliations:** 1Department of Pharmaceutical Sciences, Daniel K. Inouye College of Pharmacy, University of Hawai‘i at Hilo, Hilo, HI 96720, USA; feifeiliu@jsnu.edu.cn (F.L.); sudipta1787@gmail.com (S.M.); ruxian.love@163.com (R.R.); dateabhi@hawaii.edu (A.D.); 2School of Life Sciences, Jiangsu Normal University, Xuzhou 221116, China; 3Department of Chemistry, 2545 McCarthy Mall, University of Hawai‘i at Manoa, Honolulu, HI 96822, USA; tjod@hawaii.edu (T.J.O.); yuheng@hawaii.edu (Y.L.); ruisun@hawaii.edu (R.S.); philipwi@hawaii.edu (P.G.W.); 4Daniel K. Inouye U.S. Pacific Basin Agricultural Research Center, USDA-ARS, Hilo, HI 96720, USA; marisa.wall@usda.gov; 5Department of Pharmacy Practice, Daniel K. Inouye College of Pharmacy, University of Hawai‘i at Hilo, Hilo, HI 96720, USA; supakit@hawaii.edu; 6Department of Pharmacology and Toxicology, R. K. Coit College of Pharmacy, University of Arizona, Tucson, AZ 85715, USA; 7Hale Ola Pono, LLC, Keaau, HI 96749, USA; lomidoc@gmail.com

**Keywords:** *Waltheria indica*, coumarinolignans, antioxidant, anti-inflammatory

## Abstract

Seven new coumarinolignans, walthindicins A–F (**1a**, **1b**, **2**–**5**, **7**), along with five known analogs (**6**, **8**–**11**), were isolated from the roots of *Waltheria indica*. The structures of the new compounds are determined by detailed nuclear magnetic resonance (NMR), circular dichroism (CD) with extensive computational support, and mass spectroscopic data interpretation. Compounds were tested for their antioxidant activity in Human Cervical Cancer cells (HeLa cells). Compounds **1a** and **6** showed higher reactive oxygen species (ROS) inhibitory activity at 20 μg/mL when compared with other natural compound-based antioxidants such as ascorbic acid. Considering the role of ROS in nuclear-factor kappa B (NF-κB) activation, compounds **1a** and **6** were evaluated for NF-κB inhibitory activity and showed a concentration-dependent inhibition in Human Embryonic Kidney 293 cells (Luc-HEK-293).

## 1. Introduction

*Waltheria indica* L. (Malvaceae) is a short-lived shrub widespread in subtropical and tropical regions, including Hawai‘i [1,2]. In Hawaiian traditional medicinal practices, the indigenous ‘uhaloa, *Waltheria indica* var. *americana* is one of the most recognized plants and has been used to treat asthma, inflammation, neuralgia and pain [3,4]. ‘Uhaloa has been used in Hawaiian traditional medicine (lā‘au lapa‘au) by native healers to treat a variety of illnesses including asthma, skin inflammation, tuberculosis, tooth abscesses, as well as other infectious diseases. The root bark is used for sore throat relief, whereas tea made from the leaves helps fever and respiratory problems [5]. Furthermore, the roots, aerial parts, and whole plant are used in several countries for treating various conditions such as asthma, respiratory illness such as a sore throat and cough, inflammation, fever, pain, skin infections, and cancer [3,6].

A previous biological investigation of *W. indica* extracts showed potential anti-inflammatory activities by inhibiting the expression of key inflammatory cytokines and cytokine receptors such as interleukin (IL)-1 family: IL-1β, IL-1Ra, IL-18 and IL-6, and additionally, through the reduced expression of tumor necrosis factor alpha (TNF-α) and its receptor TNF RII, and inhibition of the mRNA and protein levels of NF-κB in human macrophages [7]. The IL-1 family is primarily associated with innate immunity [7]. Previous phytochemical studies of *W. indica* reported polyphenols [8], flavonoids [9,10,11], coumarins, terpenoids [12], cyclopeptides, and quinoline alkaloids [13,14,15]. In this context, flavonoids and quinoline alkaloids isolated from *W. indica* possess NF-κB inhibitory and quinone reductase inducing activities. [5]. There have been no phytochemical studies of coumarinolignans from roots of *W. indica* growing in Hawai‘i. In addition, coumarinolignans from *Eurycorymbus cavaleriei* L. (Sapindaceae) and *Hyoscyamus niger* L. (Solanaceae) have exhibited many interesting biological activities [16,17]. Most of the studies reported anti-inflammatory, cytotoxic, and hepatoprotective activities [16,17]. Considering these factors, we conducted this current study. 

## 2. Results and Discussion

Walthindicin A (**1**) was obtained as a pale-yellow powder. A molecular formula of C_20_H_18_O_8_ with twelve degrees of unsaturation was established on the basis of the HRESIMS (*m*/*z* 387.1081 [M + H]^+^). The IR spectrum suggested the presence of hydroxy group (3422 cm^−1^), conjugated carbonyl (1718 cm^−1^), and aromatic ring (1620 and 1515 cm^−1^) functionalities. The ^1^H NMR data (in CDCl_3_, 400 MHz, Table 1) of **1** indicated the presence of three aromatic protons (δ_H_ 6.89, d, H-2′; δ_H_ 6.95, d, H-5′; δ_H_ 6.94, dd, H-6′) attributed to a 1,3,4-trisubstituted benzene ring, one singlet proton (δ_H_ 6.50, s, H-8) of another aromatic ring, four oxymethine/oxymethylene protons (δ_H_ 3.55, dd, H-9′a; δ_H_ 3.88, m, H-9′b; δ_H_ 3.98, m, H-8′; δ_H_ 5.03, d, H-7′), two methoxy groups (δ_H_ 3.90, s, OCH_3_-3′; δ_H_ 3.92, s, OCH_3_-7), and a pair of mutually coupled vinylic protons (δ_H_ 6.17, d, H-2; δ_H_ 7.90, d, H-3). The ^13^C NMR data (in CDCl_3_, 100 MHz, Table 2) of **1** displayed 20 carbons, including two methyls (δ_C_ 56.3 and 56.7), one methylene (δ_C_ 61.6), two oxymethines (δ_C_ 77.3 and 78.5), six olefinic methines (δ_C_ 93.1, 109.9, 112.3, 115.0, 121.0 and 138.3) and nine quaternary carbons (δ_C_ 103.7, 127.4, 129.6, 140.0, 146.8, 147.1, 149.9, 152.4 and 161.7). All of the proton signals were assigned to the corresponding carbons through direct ^1^H and ^13^C correlations in the HSQC spectrum. The spectroscopic data of **1** were closely related to those of reevesiacoumarin [18], a known lignanoid with a coumarinolignan skeleton, which was previously isolated from the root and stem of *Reevesia formosana* (Malvaceae). The molecular formula of compound **1** was C_20_H_18_O_8_. Its molecular formula is different from reevesiacoumarin by 16 mass units. Analysis of the NMR data revealed that a hydroxy group in reevesiacoumarin was absent, and an additional aromatic proton (δ_H_ 6.96, d, H-5′) was present in **1** (Figure 1). This conclusion was supported by the upfield shift of the C-5′ (δ_C_ 115.0 in **1** vs. 147.1 in reevesiacoumarin) and was confirmed by the correlations of H-5′ (δ_H_ 6.96) with C-1′ (δc 127.4) and C-3′ (δc 147.1) in the HMBC spectrum (Figure 2). The C-6–O–C-8′ (C-5–O–C-7′) linkages were confirmed through HMBC correlation signals from H-9′ (δ_H_ 3.55, 3.88) to C-6 (δc 129.6), C-7′ (δc 77.3) and C-8′ (δc 78.5). The coupling constant (*J* = 8.0 Hz) between the two vicinal oxymethine protons of C-7′ and C-8′ indicated that the phenyl group and the oxygenated methylene were *trans* diaxial [18,19]. The optical rotation of **1** was approximately zero, and the CD spectrum showed no strong signals, which proved that the compound was a racemic mixture. Further analysis by HPLC using a chiral column showed two peaks (t_R_ = 11.8 and 14.3 min) with areas in a ratio of approximately 47:53, whose CD spectra, once separated, were almost mirror images (Figure 3). 

With ECD spectra in hand, we sought to assign the absolute configuration of the two compounds (**1a** and **1b**). Initially, when analyzing the experimental spectra, based on the literature precedent, it was expected that the UV absorptions at 310–340 and 275–290 nm were due to the coumarin and the substituted benzene, respectively, with interchromophoric interactions that were producing a non-degenerate exciton coupling [20]. That report had used MM2 calculations to determine the lowest energy conformers in order to interpret the ECD spectra. However, attempts to fit our experimental data to their model were generally unsuccessful. Extensive conformational analysis and thoughtful consideration of the orbitals involved for each significant excited state led us to conclude that the published model was not an appropriate approach for the coumarinolignans isolated for this paper.

Exciton coupling is predicated on two chromophores being connected through a chiral linker. The magnitude of the coupling is inversely proportional to the square of the interchromophoric distance and is maximal around a dihedral angle of 70° for the two transition moments [21]. The isolated coumarinolignans have two chromophores connected through a chiral linker, but the dihedral angle between the chromophores is generally large (in many cases close to 180°), which places them considerably far apart, leading to minimal exciton coupling.

The ECD spectra of **1a** and **1b** contains three cotton effects, which we assigned as follows. The UV transition observed at 250 nm corresponds to a π → π∗ transition of the substituted benzene ring, with the corresponding sign of the CE dependent on the orientation of benzene rings, with respect to the plane of the coumarin. This CE is largely independent from determining the CEs in the other two regions (290 nm and 305 nm). Based on our computational data, the other two UV absorbances and associated CEs correspond to different π → π∗ transitions, located mainly on the coumarin. An example of each transition for the lowest energy conformer of **1a** is shown in Figure 4. These conclusions are supported by closer inspection of the orbitals involved in each transition for all conformers with >1% Boltzmann population for **1a** and **1b**. This conclusion does contradict the published assessment that the 310–340 and 275–290 nm CE are due to the coumarin and the substituted benzene, respectively, for this set of coumarinolignans would differently inform our approach when comparing the experimental and computational data. For this comparison, the π → π∗ absorption (250 nm) on the substituted benzene is well resolved from other the transitions and can serve as a good benchmark for aligning the experimental and computational spectra, thus ensuring the validity of the comparisons. 

From the details above, the absolute configurations of the faster eluting enantiomer and the slower eluting enantiomer were assigned as 7′*R*,8′*R* (**1a**) and 7′*S*,8′*S* (**1b**), respectively, based on the ECD calculations and their comparison to the experimental data (Figure 5). The similarity score calculated for **1a** was 0.67 with the enantiomer as 0.19 and for **1b** was 0.49 with the enantiomer as 0.30. The distortions in the experimental data may be from other transitions (other π → π∗ or n → π∗) that overlap with the π → π∗ transitions identified by the computation, but do not appear as significant in the calculated data or for other reasons.

Walthindicin B (**2**) was shown to have the molecular formula, C_21_H_20_O_8_, by HRESIMS (*m*/*z* 401.1233 [M+H]^+^, calcd for C_21_H_20_O_8_, 401.1236) and ^13^C NMR data, differing from **1** by 14 mass units. The NMR spectroscopic data of **2** (Table 1 and Table 2) showed a close correlation with those of **1,** except for an additional methoxy group (δ_H_ 3.86, δ_C_ 56.6, OCH_3_-4′). HMBC data analyses revealed that the methoxy was located on C-4′ and confirmed C-6–O–C-8′ (C-5–O–C-7′) linkages. The large coupling constant (*J* = 8.0 Hz) in **2** could be caused by the inflexible *trans* stereochemistry between H-7′ and H-8′ [18,19]. The absolute configuration of **2** was determined to be 7′*R*,8*′R* by comparison of its experimental ECD spectrum (Figure 3) with the calculated data (Appendix A). 

Walthindicin C (**3**) was obtained as a pale-yellow powder. The molecular formula of **3** was determined as C_22_H_22_O_9_ from the HRESIMS ion observed at *m*/*z* 431.1341 [M + H]^+^ (calcd for C_22_H_22_O_9_, 431.1342), differing from **2** by 30 mass units. Analysis of the NMR data (Table 1 and Table 2) revealed that an aromatic proton in **2** was absent, and, instead, one additional methoxy group (δ_H_ 3.87, δ_C_ 56.5, OCH_3_-5′) was present in **3** and substituted at C-5′. This observation was supported by the downfield shift of C-5′ (δ_C_ 153.9 in **3** vs. 113.1 in **2**) and was confirmed by the HMBC correlation signal from OCH_3_-5′ to C-5′ (δc 153.9). The C-6–O–C-8′ (C-5–O–C-7′) linkages were confirmed by the HMBC correlations of H-9′ (δ_H_ 3.58 and 3.97) with C-6 (δ_C_ 129.6), C-7′ (δ_C_ 77.3) and C-8′ (δ_C_ 78.3). The *trans* diaxial orientation between H-7′ and H-8′ was inferred from their coupling constant (*J* = 8.0 Hz) [18,19]. The absolute configuration of **3** was determined to be 7′*R*,8′*R* by comparison of its experimental ECD spectrum (Figure 3) with the calculated data (Appendix A).

Walthindicin D (**4**) was obtained as a pale-yellow powder and gave a protonated molecular ion at *m*/*z* 401.1231 [M + H]^+^ (calcd for C_21_H_21_O_8_, 401.1236) in the HRESIMS, corresponding to a molecular formula of C_21_H_20_O_8_. The ^1^H and ^13^C NMR spectra of **4** were closely related to those of compound **3**. Analysis of the NMR data (Table 1 and Table 2) revealed that one methoxy group in **3** (δ_H_ 3.86, δ_C_ 61.1, OCH_3_-4′) was absent, and an oxymethylene group (δ_H_ 3.58, 3.97, δ_C_ 61.5, OCH_2_-9′) in **3** was replaced by a methyl (δ_H_ 1.22, δ_C_ 17.5, CH_3_-9′) in **4**. The HMBC correlations from CH_3_-9′ to C-6 (*δ*_C_ 131.2), C-7′ (δ_C_ 83.4) and C-8′ (δ_C_ 75.4) in conjunction with ^1^H-^1^H COSY correlation between H-8′ (δ_H_ 4.19) and CH_3_-9′ revealed that the methyl was located on C-8′ and a C-6–O–C-8′ (C-5–O–C-7′) linkage existed. The *trans*-form between H-7′ and H-8′ was deduced from their coupling constant (*J* = 8.0 Hz) [18,19]. 

Walthindicin E (**5**) was obtained as a pale-yellow powder. Its molecular formula was determined as C_23_H_22_O_10_ with thirteen degrees of unsaturation on the basis of HRESIMS and ^13^C NMR data. The UV, IR and 1D NMR spectroscopic data of **5** were very close to those of durantin C (**6**) [22], which was purified from same subfraction by HPLC. Fortunately, there existed a pattern that C-7 (δ_C_ 137.4 in **6** vs. 136.8 in **5**) was shifted to high field and C-8 (δ_C_ 132.1 in **6** vs. 132.7 in **5**) shifted to low field when the connection mode converted from C-7–O–C-7′ to C-7–O–C-8′ [23,24]. In view of these similarities, **5** was logically assumed to be the regioisomer of **6**, which was further verified by the HMBC couplings between C-8 and H-7′ (δ_H_ 4.91), and between C-7 and H-9′ (δ_H_ 4.07) (Figure 2). The large coupling constant (*J* = 7.4 Hz) between H-7′ and H-8′ suggested that these protons were in a *trans* configuration [18,19].

The zero optical rotation suggested that **4** and **5** occurred as a racemic mixture, while compounds **4** and **5** were not further purified due to the limited quantities available of these compounds [25].

Walthindicin F (**7**) was obtained as a pale-yellow powder. The molecular formulas of **7** were determined as C_21_H_20_O_8_ from the HRESIMS ion observed at *m*/*z* 401.1235 [M+H]^+^ (calcd for C_21_H_21_O_8_, 401.1236). The UV, IR and 1D NMR spectroscopic data of **7** were very close to those of cleomiscosin A methyl ether (**8**) [17]. The same molecular formula and detailed 2D NMR analysis revealed that compound **7** shared an identical planar structure as that of **8**. Compared to **8**, the coupling constant between H-7′ and H-8′ was changed (*J* = 2.9 Hz in **7** vs. *J* = 8.0 Hz in **8**), which indicated that the relative conformation of H-7′ and H-8′ was transformed from *trans* in **8** to *cis* in **7** [18,19,26,27]. The C-7–O–C-7′ and C-8–O–C-8′ linkages were confirmed by the HMBC correlations between C-7 and H-7′ (δ_H_ 4.91), and between C-8 and H-8′ (δ_H_ 4.07). 

In terms of ECD analysis, the lowest energy conformer (Figure 6) of **7** (29.5% of the population) is representative of the majority of conformers that are present in a least > 1% abundance**.** The angle in this case is approximately 170°. The only important exceptions are conformers 3 (Figure 6) and 4 (combined 12.9% total of the Boltzmann population of **7**), which have angles around 100° between the chromophores, similar to that previously reported [20]. This angle is closer to optimal than what was observed for the other compounds isolated here and would potentially explain the stronger exciton coupling observed in this case. In fact, the calculations do show a greater involvement of the π orbitals across the molecule in the 310 nm and 250 nm transitions, indicating there is greater coupling for those conformers (Appendix A), which may indeed elicit a stronger exciton coupling for the ECD spectrum of **7**.

The calculated ECD spectrum of 7′S,8′R showed a positive Cotton effect around 245 nm and a negative Cotton effect around 320 nm, similar to the experimentally recorded CD spectrum of **7** with a similarity score of 0.80, as determined by SpecDis [28] (Figure 7). Hence, the absolute configuration of **7** was unambiguously assigned as shown.

The other five isolates obtained were identified as the known compounds durantin C (**6**) [22], cleomiscosin A methyl ether (**8**) [17], jatrocin B (**9**) [29,30], venkatasin (**10**) [20], and 4′-O-methyl-cleomiscosin D (**11**) [31], by comparison of their physical and spectroscopic data with those reported in the literature. To date, only one coumarinolignan has been reported from the Malvaceae family [18]. Our study is the first report of coumarinolignans from a plant in the *Waltheria* genus.

It is widely accepted that chronic inflammation plays a vital role in metabolic disorders such as type 2 diabetes, cardiovascular diseases, atherosclerosis and cancer [32,33]. Studies suggest that NF-κB acts as a mediator in inflammation-induced pathological conditions [34]. Significantly, the activation of NF-κB leads to the up-regulation of pro-inflammatory genes such as cytokines, chemokines, and adhesion molecules, which are chemical messengers for pathogenesis [35]. Therefore, the regulation of the inhibition of NF-κB expression has been a central target for developing new anti-inflammatory drugs. Furthermore, reactive oxygen species (ROS) play a crucial role in NF-κB activation and, thus, assessing ROS level is also an indication of pro-inflammatory responses [36]. Plant-based natural compounds/extracts have long been studied for their antioxidant and anti-inflammatory activities. Therefore, we assessed the cytotoxic concentration of all the compounds for further bioassays. ROS inhibition of compounds was tested at 20 μg/mL concentration along with other plant-based (resveratrol and ascorbic acid) and synthetic (N-acetylcysteine) positive controls (Figure 8). Compounds **1a** and **6** have shown significantly higher ROS inhibition compared to ascorbic acid and comparable activity with resveratrol. Considering the antioxidant activity of compounds **1a** and **6**, their anti-inflammatory activity was evaluated in Luc-HEK-293 cells. Tumor necrosis factor alpha (TNF-α)-induced NF-κB expression was assessed at different concentrations of compounds along with the positive control *Tosyl* phenylalanyl *chloromethyl ketone* (TPCK). Although the positive control was most potent in inhibiting NF-κB expression, 30 μg/mL of compounds **1a** and **6** showed complete inhibition of TNF-α-induced NF-κB expression and a further reduction in NF-κB expression was observed at 40 μg/mL when compared to the only cell (Figure 9).

The results suggested that tested compounds can reduce ROS production and eventually affect NF-κB activation [36]. The natural coumarinolignans are the phenylpropanoid linked via a 1,4-dioxane bridge with the coumarin skeleton. These closely related coumarinolignans show promising biological activity. The fusion between the 1,4-dioxane bridge and coumarin may occur either at the 5 and 6 positions or at the 7 and 8 positions. Compared with **1a** and **1b**, chiral centers (7′*R*,8′*R*) play an important role in our ROS inhibition assay. When comparing compounds **1a** and **6**, structural differences presumably act through a different cellular mechanism that may affect ROS production and affect NF-κB activation. Furthermore, the fusion of a phenylpropanoid with two ortho-hydroxy groups may take place in two different manners giving rise to regioisomers (**5** and **6**). The C-7–O–C-8′ linked (**6**) showed better ROS inhibition activity than the C-7–O–C-7′ (**5**). The hydroxy group at C-4′ also contributes to the ROS inhibitory activity. However, the inhibition of ROS production by these compounds further confirms the potential of these compounds as chemo-preventive agents. 

## 3. Materials and Methods

### 3.1. General Experimental Procedures

Optical rotations were recorded in CHCl_3_ on a Rudolph Research AUTOPOL IV multiwavelength polarimeter (Rudolph Research Analytical, Hackettstown, NJ, USA). Ultraviolet spectra were measured with a Shimadzu PharmaSpec-1800 UV-visible spectrophotometer (Shimadzu Scientific Instruments, Columbia, MD, USA). Electronic circular dichroism spectra were obtained at 20 °C on a JASCO J-815 spectropolarimeter (JASCO Inc., Tokyo, Japan). Infrared radiation spectra were recorded on a Thermo Scientific Nicolet iS 10 FT-IR spectrometer (Thermo Fisher Scientific, Waltham, MA, USA). One-dimensional- and two-dimensional-NMR spectra were collected on a Bruker AVANCE DRX-400 NMR spectrometer (Bruker, Billerica, MA, USA) at 400 MHz for ^1^H and 100 MHz for ^13^C, and the data were processed using TopSpin 3.2 software with CDCl_3_ (δ_H_ 7.23, δ_C_ 77.16) or CD_3_OD (δ_H_ 3.31, δ_C_ 49.0) as solvents. High-resolution electrospray ionization mass spectra were performed with an Agilent 6530 LC-qTOF High Mass Accuracy mass spectrometer (Santa Clara, CA, USA) under the positive-ion mode. Silica gel (230–400 mesh, 480–800 mesh, Sorbent Technologies, Atlanta, GA, USA), Sephadex LH-20 (GE Healthcare, Piscataway, NJ, USA), and MCI gel (CHP-20P, Mitsubishi Chemical Corporation, Tokyo, Japan) were used for column chromatography. Preparative HPLC was performed on a Thermo Scientific Ultimate 3000 system equipped with a photodiode array detector, using a YMC reversed-phase C_18_ column (5 μm, 20 × 250 mm, YMC-pack ODS-A) with a flow rate of 5 mL/min or a reversed-phase C_18_ chiral column (250 × 10 mm, 5 μm, Cellulose-1), with a flow rate of 4 mL/min. 

### 3.2. Plant Material

The fresh roots of *W. indica* were collected from Puako, Hawai‘i Island (Big Island), Hawaii, USA, in November 2019 and were identified by Kumu Dane Kaohelani Silva. A voucher specimen (No. WIS01) was deposited at the Natural Product Chemistry Laboratory, Daniel K. Inouye College of Pharmacy, University of Hawai‘i at Hilo.

### 3.3. Extraction and Isolation

The air-dried powdered roots (11.5 kg) of *W. indica* were extracted with methanol (60 L × 3) for 48 h at room temperature. The methanolic extract was filtered and concentrated under reduced pressure, to afford a crude extract (1620 g), and then successively extracted with *n*-hexane, ethyl acetate, and *n*-butanol. After solvent removal, the ethyl acetate-soluble partition (105.0 g) was purified on an MCI gel CHP-20P column, eluted with H_2_O-MeOH (1:0 to 0:1, *v*/*v*), and finally with acetone, to yield seven fractions. The 80% MeOH fraction (13.0 g) was chromatographed over a silica gel column (*n*-hexane-EtOAc, 100:0 to 0:100, and finally CHCl_3_-MeOH, 1:1) to yield 18 fractions (Fr. 1–18), which were combined on the basis of thin layer chromatography analysis. Fr. 4 (520.0 mg) was applied to a Sephadex LH-20 column eluting with MeOH to afford three fractions (4.1–4.3). Fraction 4.2 (31.0 mg) was separated on a silica gel column (CHCl_3_-MeOH, 200:1 to 50:1) and further purified by semipreparative HPLC (acetonitrile-H_2_O, 35:65 to 75:25) to yield compound **4** (2.1 mg). Fr. 5 (650.0 mg) was loaded on a Sephadex LH-20 column with MeOH as eluent to give two fractions (5.1–5.2). Fraction 5.1 (500.0 mg) was chromatographed on a silica gel column (*n*-hexane-acetone, 4:1 to 2:1) to yield five subfractions (5.1.1–5.1.5). Subfraction 5.1.5 (50.0 mg) was further purified on a silica gel column (CHCl_3_-acetone, 500:1) to yield compounds **9** (19.6 mg) and **10** (2.7 mg). Fr. 6 (1.84 g) was separated on a silica gel column (CH_3_Cl-acetone, 20:1 to 1:1) to obtain thirteen fractions (6.1-6.13). Fraction 6.1 (69.0 mg) was separated on a silica gel column (*n*-hexane-acetone, 2:1) and further purified by preparative HPLC (MeOH-H_2_O, 60:40 to 75:25) to furnish compounds **5** (2.0 mg) and **6** (14.3 mg). Fraction 6.3 (40.0 mg) was subjected to a silica gel column (CH_3_Cl-EtOAc, 10:1) to obtain compounds **2** (15.8 mg) and **3** (8.8 mg). Fraction 6.4 (118.0 mg) was chromatographed on a silica gel column (*n*-hexane-acetone, 4:1 to 1:1) to yield nine subfractions (6.4.1–6.4.9). Subfraction 6.4.5 (22.4 mg) was subjected to a silica gel column (CHCl_3_-acetone, 40:1 and 15:1) to yield compounds **8** (10.5 mg) and **11** (4.5 mg). Subfraction 6.4.6 (10.7 mg) was further purified on a silica gel column (*n*-hexane-isopropanol, 3:1) to yield compound **7** (4.2 mg). Fraction 6.6 (92.0 mg) was chromatographed on a silica gel column (*n*-hexane-acetone, 8:1 to 2:1) to yield compound **1** (12.5 mg). Compound **1** was further purified by preparative chiral HPLC (acetonitrile-H_2_O, 40:60) to furnish compounds **1a** (3.4 mg) and **1b** (3.8 mg).

Compound **1a**: pale-yellow powder; ${\left[\alpha \right]}_{D}^{20}$ −43 (*c* 0.13, CHCl_3_); UV_max_ (CHCl_3_) *λ*_max_ (log *ε*) 320 (4.26), 288 (4.03), 245 (4.33) nm; IR (film) *ν*_max_ 3422, 2955, 2920, 2850, 1718, 1620, 1565, 1515, 1457, 1373, 1266, 1202, 1150, 1121, 1092, 1034, 825, 758 cm^−1^; CD (CHCl_3_) λ_max_ (Δε) 257 (−0.44), 274 (+0.51), 307 (−0.22), 336(+0.18); ^1^H NMR date, see Table 1; ^13^C NMR data, see Table 2; HRESIMS *m*/*z* 387.1081 [M + H]^+^ (calcd for C_20_H_19_O_8_, 387.1080).

Compound **1b**: pale-yellow powder; ${\left[\alpha \right]}_{D}^{20}$ 49 (*c* 0.24, CHCl_3_); UV_max_ (CHCl_3_) *λ*_max_ (log *ε*) 320 (4.15), 288 (4.06), 245 (4.42) nm; IR (film) *ν*_max_ 3421, 2959, 2925, 2853, 1720, 1617, 1565, 1514, 1454, 1368, 1270, 1204, 1150, 1121, 1087, 1032, 826, 757 cm^−1^; CD (CHCl_3_) λ_max_ (Δε) 257 (+1.39), 294 (+1.97), 330 (−0.78); ^1^H NMR date, see Table 1; ^13^C NMR data, see Table 2; HRESIMS *m*/*z* 387.1080 [M + H]^+^ (calcd for C_20_H_19_O_8_, 387.1080).
Compound **2**: pale-yellow powder; ${\left[\alpha \right]}_{D}^{20}$ −44 (*c* 0.53, CHCl_3_); UV_max_ (CHCl_3_) *λ*_max_ (log *ε*) 320 (4.30), 288 (4.12), 245 (4.43) nm; IR (film) *ν*_max_ 3459, 3014, 2960, 2922, 2845, 1720, 1617, 1564, 1517, 1455, 1366, 1257, 1142, 1118, 1089, 1018, 858, 805, 752 cm^−1^; CD (CHCl_3_) λ_max_ (Δε) 242 (−3.63), 316 (−1.06); ^1^H NMR date, see Table 1; ^13^C NMR data, see Table 2; HRESIMS *m*/*z* 401.1233 [M + H]^+^ (calcd for C_21_H_21_O_8_, 401.1236).

Compound **3**: pale-yellow powder; ${\left[\alpha \right]}_{D}^{20}$ −53 (*c* 0.29, CHCl_3_); UV_max_ (CHCl_3_) *λ*_max_ (log *ε*) 320 (4.30), 245 (4.43) nm; IR (film) *ν*_max_ 3458, 3008, 2962, 2925, 2841, 1726, 1622, 1590, 1567, 1509, 1452, 1426, 1368, 1342, 1241, 1143, 1120, 1086, 1034, 803, 754 cm^−1^; CD (CHCl_3_) λ_max_ (Δε) 242 (−4.80), 338 (−1.22); ^1^H NMR date, see Table 1; ^13^C NMR data, see Table 2; HRESIMS *m*/*z* 431.1341 [M + H]^+^ (calcd for C_22_H_23_O_9_, 431.1342).

Compound **4**: pale-yellow powder; ${\left[\alpha \right]}_{D}^{20}$ 0 (*c* 0.07, CHCl_3_); UV_max_ (CHCl_3_) *λ*_max_ (log *ε*) 320 (3.88), 245 (4.04) nm; IR (film) *ν*_max_ 3427, 3013, 2958, 2924, 2852, 1721, 1620, 1566, 1517, 1450, 1367, 1341, 1255, 1209, 1149, 1114, 1085, 1033, 826, 757 cm^−1^; ^1^H NMR date, see Table 1; ^13^C NMR data, see Table 2; HRESIMS *m*/*z* 401.1231 [M + H]^+^ (calcd for C_21_H_21_O_8_, 401.1236).

Compound **5**: pale-yellow powder; ${\left[\alpha \right]}_{D}^{20}$ 0 (*c* 0.13, CHCl_3_); UV_max_ (CHCl_3_) *λ*_max_ (log *ε*) 318 (3.81), 245 (4.04) nm; IR (film) *ν*_max_ 3418, 3012, 2957, 2923, 2854, 1719, 1612, 1575, 1500, 1454, 1411, 1365, 1301, 1221, 1154, 1134, 1111, 1051, 835, 751 cm^−1^; ^1^H NMR date, see Table 1; ^13^C NMR data, see Table 2; HRESIMS *m*/*z* 459.1289 [M + H]^+^ (calcd for C_23_H_23_O_10_, 459.1291).

Compound **7:** pale-yellow powder; ${\left[\alpha \right]}_{D}^{20}$ −120 (c 0.14, CHCl_3_); UV_max_ (CHCl_3_) λ_max_ (log ε) 320 (3.85), 287 (3.67), 245 (4.06) nm; IR (film) ν_max_ 3468, 3009, 2963, 2928, 2854, 1722, 1613, 1573, 1517, 1498, 1443, 1415, 1300, 1254, 1239, 1130, 1056, 1027, 837, 763 cm^−1^; CD (CHCl_3_) λ_max_ (Δε) 246 (+2.55), 274 (+1.47), 316 (−4.97); ^1^H NMR date, see Table 1; ^13^C NMR data, see Table 2; HRESIMS *m*/*z* 401.1235 [M + H]^+^ (calcd for C_21_H_21_O_8_, 401.1236). 

### 3.4. Computational Methods

For ECD prediction, conformers within 5 kcal/mol of the lowest energy conformer were searched using the Monte Carlo multiple minimum (MCMM) method [37] and the OPLS-2005 force field [38] in Schrodinger Inc.’s MacroModel [39] and then optimized in Gaussian 09 [40] at the CAM-B3LYP [41]/6-31+G(d,p) [42,43,44,45] level with a Polarizable Continuum Model (PCM) [46] in chloroform. The optimized conformers were subsequently verified by frequency calculations at the same level. The geometries of all conformers close in energy were checked for redundancy. For CD prediction, time-dependent density functional theory (TDDFT) [47,48] were conducted at the CAM-B3LYP [42,49]/def2-TZVPP [50] level to calculate the electronic excitation energies and rotational strengths with PCM in chloroform. Boltzmann weighted ECD spectra, where conformers with >1% Boltzmann population were calculated using SpecDis [28] for comparison by similarity factor with the experimentally determined data recorded in chloroform. The most up-to-date version (as of October 2021) of Multiwfn [51] software was used for visualization of the molecular orbitals (isovalue = 0.03) involved in UV and ECD transitions. 

### 3.5. Assessment of ROS Inhibition in HeLa Cells 

HeLa cells were seeded in a white-walled 96-well plate at 20 × 10^3^ cells per well and maintained in 200 μL media. Cells were incubated for 48 h at 37 °C in 5% CO_2_, and then, the medium was replaced with fresh (190 µL media + 10 µL sample) media and incubated for 6 h. Sample stock was prepared in DMSO and diluted with PBS. Final concentration of tested samples was 20 µg/mL along with positive control Ascorbic acid (Sigma Aldrich, Allentown, PA, USA), resveratrol (ACROS Organics) and *N*-acetyl-*l*-Cysteine (ACROS Organics). Next, 5 µL (0.5 ng/well) of diluted TNF-α solution was added to each well. The plate was incubated for 5–6 h. The cells were washed in buffer and stained with 2′,7′–dichlorofluorescin diacetate (DCFDA, Abcam, Boston, MA, USA) for 45 min. A measurement was taken by a microplate reader (Synergy H1, BioTek, Winooski, VT, USA) with excitation/emission at 485 nm/535 nm. All samples were tested in triplicates. Experiments were conducted in triplicates to ensure the reproducibility of the results. 

### 3.6. TNF-α Activated NF-κB Assay

Human embryonic kidney cells 293 (Panomic, Fremont, CA, USA) were employed for monitoring changes occurring in the NF-κB pathway [52]. Stable constructed cells were seeded into 96-well plates at 20 × 10^3^ cells per well. Cells were maintained in Dulbecco’s modified Eagle’s medium (Invitrogen Co., Carlsbad, CA, USA), supplemented with 10% fetal bovine serum (FBS), 100 units/mL penicillin, 100 μg/mL streptomycin, and 2 mM L-glutamine. After 48 h of incubation, the medium was replaced with a fresh media containing different concentration of compounds (**1a** and **6**) along with *Tosyl* phenylalanyl *chloromethyl ketone* (TPCK) as a positive control. TNF-*α* (recombinant, human, *E. coli*; Calbiochem, Gibbstown, NJ, USA) was used as an activator at a concentration of 2 ng/mL (0.14 nM). After 6 h incubated, the spent medium was discarded, and the cells were washed once with PBS. Cells were lysed using 50 μL (for 96-well plate) of Reporter Lysis Buffer from Promega, by incubating for 5 min on a shaker. The luciferase assay was performed using the Luc assay system (Promega). The gene product, luciferase enzyme, reacted with the luciferase substrate, emitting light that was detected using a luminometer (BioTek Synergy H1 Hybrid Multi-mode Microplate Reader, Winooski, VT, USA). All samples were tested in triplicates. Experiments were conducted in triplicates to ensure the reproducibility of the results. 

### 3.7. Statistical Analysis

Data were analyzed using the GraphPad Prism version 8.0. Descriptive statistics were calculated for all relevant variables. One-way ANOVA with Tukey’s HSD test was used to compare the means of ROS and NF-κB inhibitory activity among and between compounds. The level of significance for all analyses was set at an alpha equal to 0.05.

## 4. Conclusions

A chemical investigation of the root of *W. indica* from Big Island, Hawaii, United States, yielded seven new coumarinolignans, walthindicins A–F (**1a**, **1b**, **2**–**5**, **7**), together with five known compounds (**6**, **8**–**11**). Except for compound **7**, other compounds share the *trans* diaxial orientation between H-7′ and H-8′. Compounds **1a** and **6** showed superior ROS inhibitory activity at 20 μg/mL in HeLa cells when compared with the positive control ascorbic acid. Compounds **1a** and **6** showed moderate NF-κB inhibitory activity in a concentration-dependent manner in Luc-HEK-293 cells. Our findings identify natural products with antioxidant properties and provide evidence for the application of *W. indica* in the treatment of inflammatory-related diseases.

## Figures and Tables

**Figure 1 molecules-27-03270-f001:**
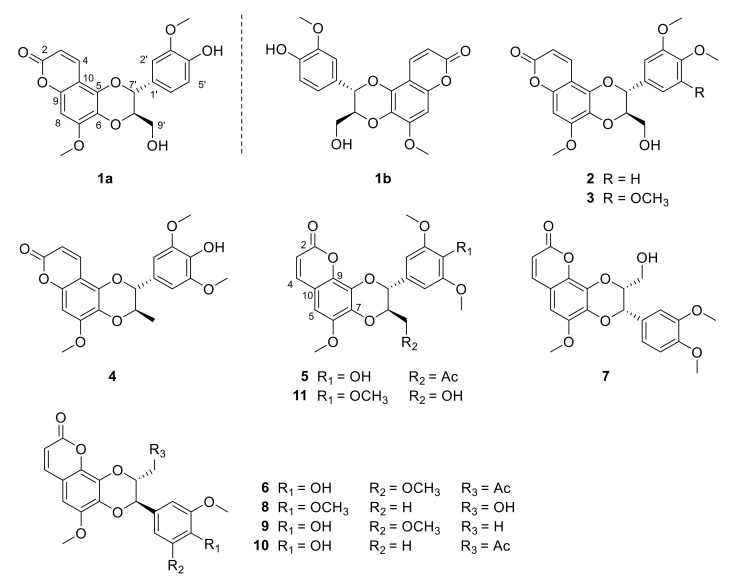
Structures of compounds **1a**–**11** from the roots of *Waltheria indica*.

**Figure 2 molecules-27-03270-f002:**
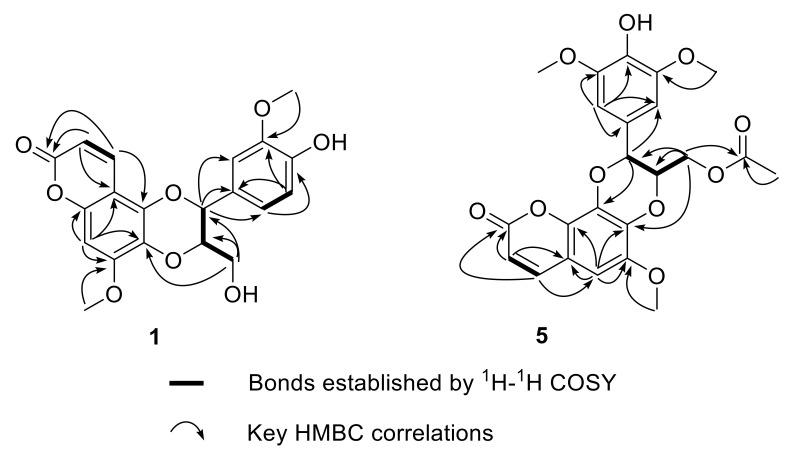
Key ^1^H-^1^H COSY and HMBC correlations of compounds **1** and **5**.

**Figure 3 molecules-27-03270-f003:**
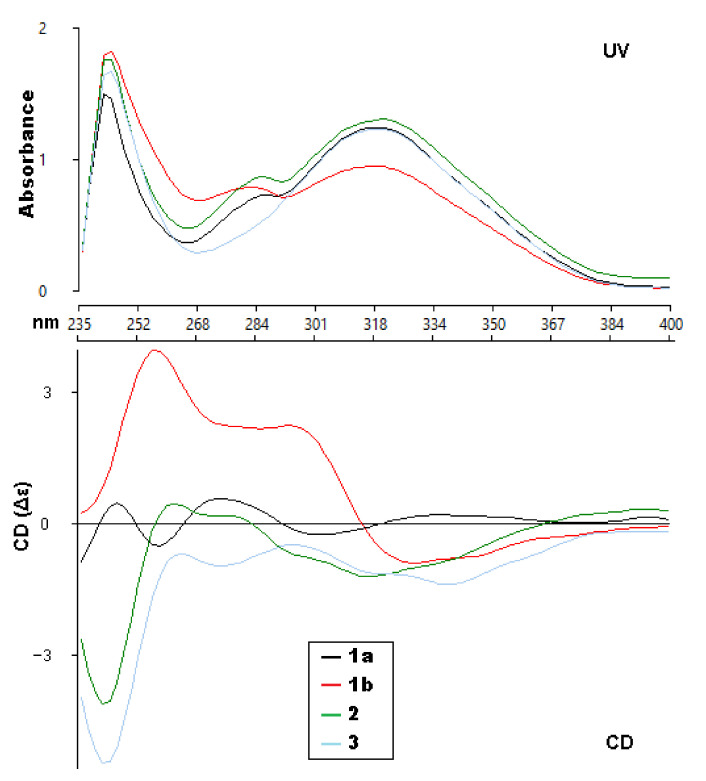
Experimental UV and ECD spectra (CHCl_3_) of compounds **1a**, **1b**, **2** and **3**.

**Figure 4 molecules-27-03270-f004:**
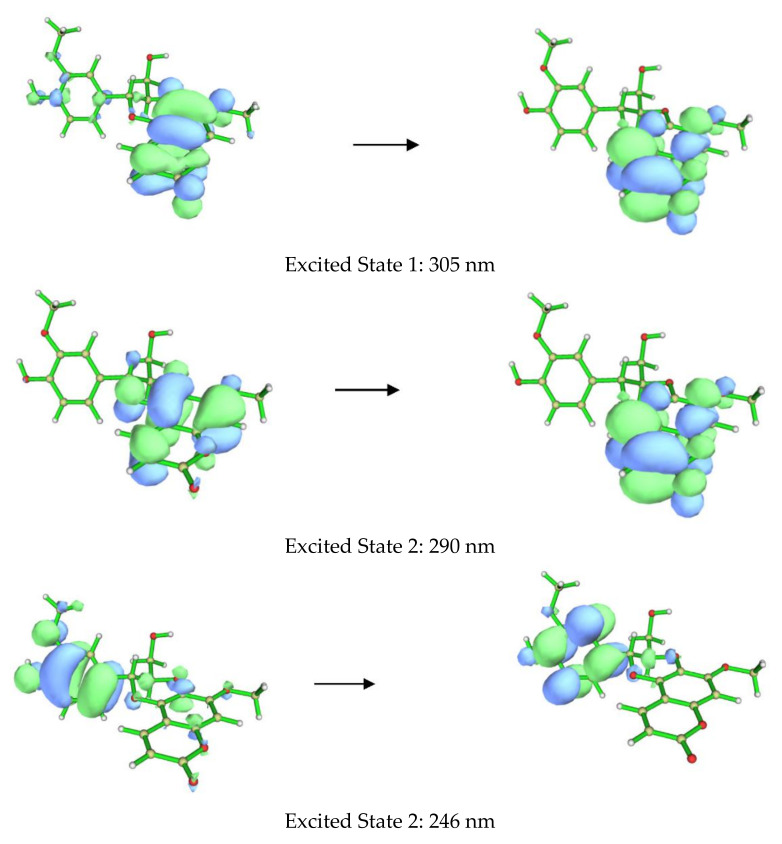
Orbitals involved in the excited states for the lowest energy conformer of **1a**.

**Figure 5 molecules-27-03270-f005:**
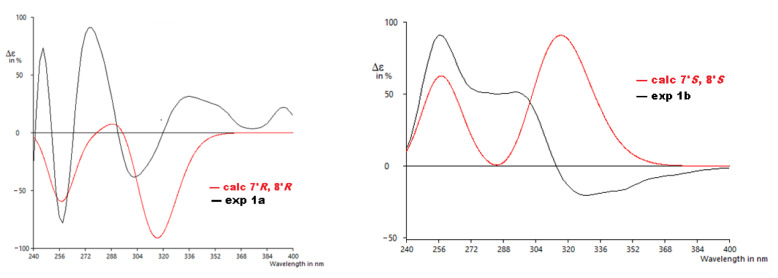
Experimental ECD to calculated ECD spectra for **1a** and **1b**.

**Figure 6 molecules-27-03270-f006:**
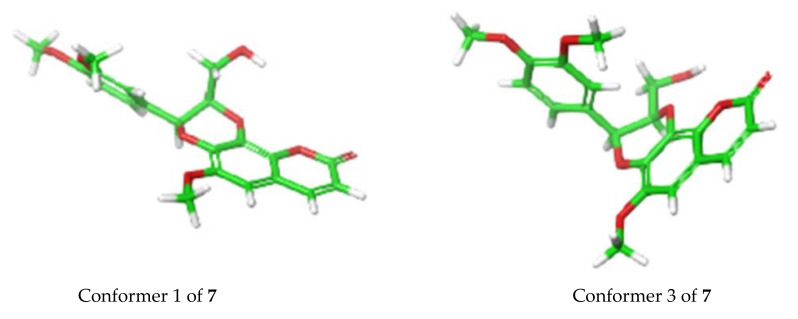
Conformer 1 (29.5%) vs. conformer 3 (8.9%) for **7** determined by TDDFT calculations.

**Figure 7 molecules-27-03270-f007:**
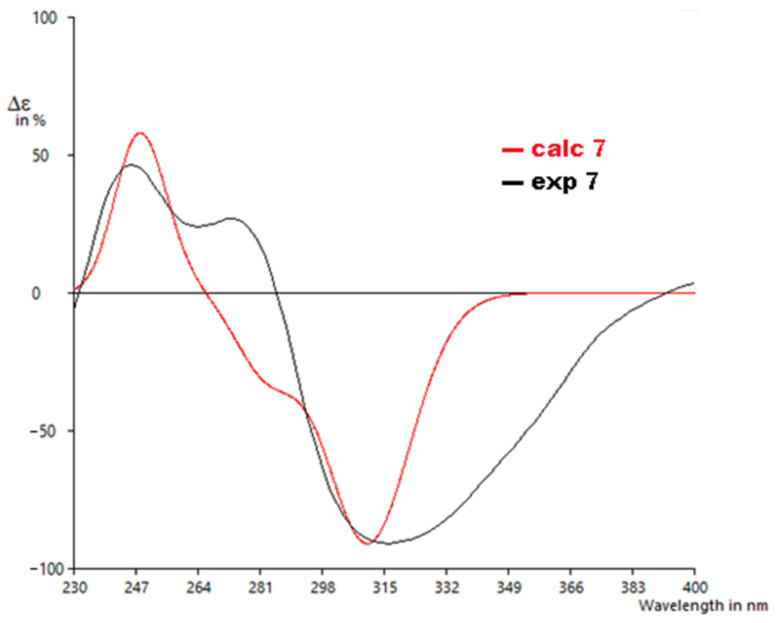
Experimental to calculated ECD spectra for **7**.

**Figure 8 molecules-27-03270-f008:**
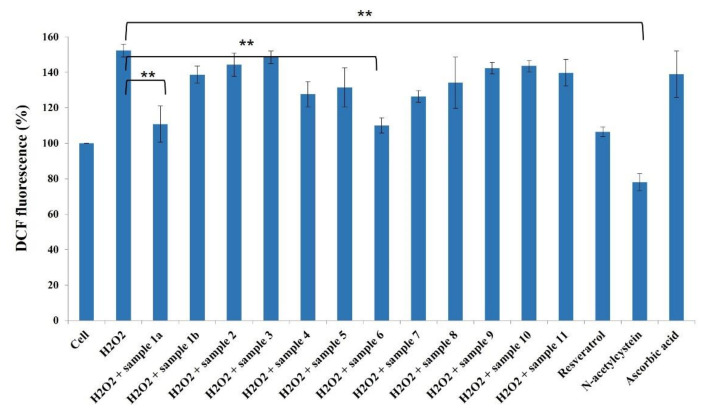
Effects of compounds **1a**–**11** (20 µg/mL) against ROS production in HeLa cells. Results are expressed as mean ± SD., ** *p* < 0.01 control group Cell + H_2_O_2_ versus samples.

**Figure 9 molecules-27-03270-f009:**
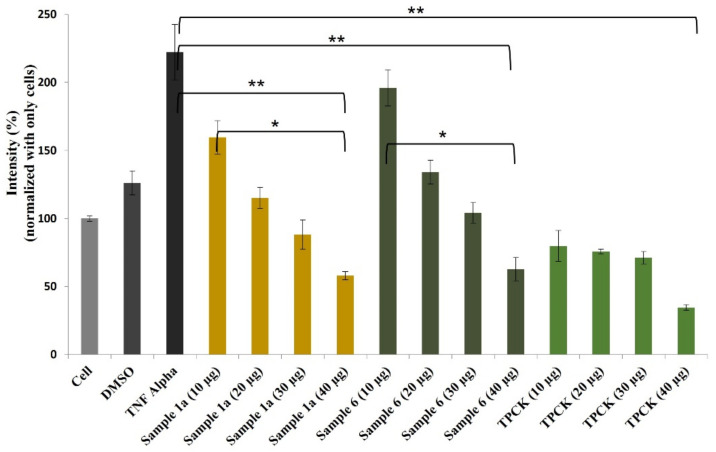
Effects of compounds **1a** and **6** against NF-κB production in Luc-HEK-293 cells. Results are expressed as mean ± SD. * *p* < 0.05, ** *p* < 0.01 control group Cell + TNF alpha versus samples.

**Table 1 molecules-27-03270-t001:** ^1^H NMR (400 MHz) Data for Compounds **1a**–**5** and **7** (*δ* in ppm, *J* in Hz).

Position	1a ^a^	1b ^a^	2 ^b^	3 ^a^	4 ^b^	5 ^a^	7 ^a^
3	6.17, d (9.6)	6.18, d (9.6)	6.16, d (9.6)	6.18, d (9.6)	6.16, d (9.6)	6.28, d (9.5)	6.29, d (9.5)
4	7.90, d (9.6)	7.90, d (9.6)	7.96, d (9.6)	7.92, d (9.6)	7.98, d (9.6)	7.58, d (9.5)	7.59, d (9.5)
5						6.53, (s)	6.53, (s)
8	6.50, (s)	6.51, (s)	6.62, (s)	6.52, (s)	6.63, (s)		
2′	6.89, d (1.5)	6.90, (brs)	7.06, d (1.4)	6.65, (s)	6.74, (s)	6.58, (s)	6.86, d (2.0)
5′	6.95, d (8.0)	6.96, d (8.2)	7.01, d (8.2)				6.81, d (8.3)
6′	6.94, dd (1.5, 8.0)	6.95, d (8.2)	7.03, dd (1.4, 8.2)	6.65, (s)	6.74, (s)	6.58, (s)	6.92, dd (2.0, 8.3)
7′	5.03, d (8.0)	5.04, d (8.0)	5.07, d (8.0)	5.05, d (8.0)	4.70, d (8.0)	4.91, d (7.4)	5.34, d (2.9)
8′	3.98, ddd (2.9, 3.5, 8.0)	3.99, (m)	4.08, ddd (2.4, 4.0, 8.0)	4.00, (m)	4.19, dq (8.0, 6.4)	4.36, (m)	4.63, ddd (2.9, 4.0, 8.3)
9′	3.55, dd (3.5, 12.8) 3.88, dd (2.9, 12.8)	3.56, dd (3.4, 12.5) 3.89, (m)	3.49, dd (4.0, 12.5) 3.78, dd (2.4, 12.5)	3.58, dd (3.0, 12.5) 3.97, (m)	1.22, d (6.4)	4.07, dd (5.4, 13.3) 4.35, (m)	3.57, dd (4.0, 12.3) 3.79, dd (8.3, 12.3)
Ac-9′						2.05 (s)	
OCH_3_-5						3.90 (s)	3.87 (s)
OCH_3_-7	3.92 (s)	3.93 (s)	3.93 (s)	3.93 (s)	3.93 (s)		
OCH_3_-3′	3.90 (s)	3.91 (s)	3.85 (s)	3.87 (s)	3.87 (s)	3.87 (s)	3.82 (s)
OCH_3_-4′			3.86 (s)	3.86 (s)			3.83 (s)
OCH_3_-5′				3.87 (s)	3.87 (s)	3.87 (s)	

^a^ Compounds **1a**, **1b**, **3**, **5** and **7** were measured in CDCl_3_. ^b^ Compounds **2** and **4** were measured in MeOD.

**Table 2 molecules-27-03270-t002:** ^13^C NMR (100 MHz) Data for Compounds **1a**–**5** and **7** (*δ*_C_ in ppm, type).

Position	1a ^a^	1b ^a^	2 ^b^	3 ^a^	4 ^b^	5 ^a^	7 ^a^
2	161.7, C	161.7, C	163.8, C	161.6, C	163.9, C	160.8, C	160.9, C
3	112.3, CH	112.3, CH	112.4, CH	112.4, CH	112.3, CH	114.6, CH	114.5, CH
4	138.3 CH	138.3 CH	140.0, CH	138.1, CH	140.2, CH	143.8, CH	144.0, CH
5	140.0, C	140.0, C	141.4, C	139.8, C	141.5, C	100.9, CH	101.0, CH
6	129.6, C	129.6, C	131.4, C	129.6, C	131.2, C	146.0, C	146.1, C
7	152.4, C	152.4, C	154.4, C	152.4, C	154.3, C	136.8, C	137.0, C
8	93.1, CH	93.1, CH	94.0, CH	93.2, CH	93.9, CH	132.7, C	130.9, C
9	149.9, C	149.9, C	150.9, C	149.9, C	150.8, C	139.2, C	139.3, C
10	103.7, C	103.7, C	104.7, C	103.7, C	104.7, C	112.0, C	112.0, C
1′	127.4, C	127.4, C	130.1, C	131.1, C	128.4, C	125.8, C	127.6, C
2′	109.9, CH	109.8, CH	112.5, CH	104.6, CH	106.3, CH	104.6, CH	110.2, CH
3′	147.1, C	147.1, C	150.9, C	153.9, C	149.7, C	147.6, C	149.4, C
4′	146.8, C	146.8, C	151.4, C	138.9, C	137.8, C	136.1, C	149.7, C
5′	115.0, CH	115.0, CH	113.1, CH	153.9, C	149.7, C	147.6, C	111.6, CH
6′	121.0, CH	121.0, CH	121.8, CH	104.6, CH	106.3, CH	104.6, CH	119.4, CH
7′	77.3, CH	77.3, CH	78.6, CH	77.3, CH	83.4, CH	76.7, CH	76.4, CH
8′	78.5, CH	78.5, CH	79.8, CH	78.3, CH	75.4, CH	76.2, CH	77.6, CH
9′	61.6, CH_2_	61.6, CH_2_	62.0, CH_2_	61.5, CH_2_	17.5, CH_3_	63.0, CH_2_	60.0, CH_2_
Ac-9′						170.6, C	
Ac-9′						21.0, CH_3_	
OCH_3_-5						56.7, CH_3_	56.6, CH_3_
OCH_3_-7	56.7, CH_3_	56.5, CH_3_	57.2, CH_3_	56.7, CH_3_	57.1, CH_3_		
OCH_3_-3′	56.3, CH_3_	56.3, CH_3_	56.7, CH_3_	56.5, CH_3_	57.1, CH_3_	56.7, CH_3_	56.3, CH_3_
OCH_3_-4′			56.6, CH_3_	61.1, CH_3_			56.1, CH_3_
OCH_3_-5′				56.5, CH_3_	57.1, CH_3_	56.7, CH_3_	

^a^ Compounds **1a**, **1b**, **3**, **5** and **7** were measured in CDCl_3_. ^b^ Compounds **2** and **4** were measured in MeOD.

## Data Availability

Not applicable.

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
