# Peer review of "Coumarinolignans with Reactive Oxygen Species (ROS) and NF-κB Inhibitory Activities from the Roots of Waltheria indica"

_molecules, 2022, doi:10.3390/molecules27103270_

Round 1

Reviewer 1 Report

Manuscript is well and clearly written. The subject is interesting and worthy of investigation. The detailed documentation of the results has been provided. I have only few suggestions for Authors.

  • A few sentences on biological activity of coumarinolignans could be added to Introduction
  • Table 1 should be mention in text before Table 2 – or change numbering of Table 1 and 2
  • Page 3: Structures should be described as Figure 1 – lack of legend for this figure.
  • Page 8 „which was purified from same subfraction by HPLC using a column” – the word „column” is unnecessary because HPLC always uses column
  • 3. Extraction and Isolation – the scheme of experimental procedure (given in Supplementary material) would make easier following
  • Some editorial errors, e.g. lack of space between value and unit (e.g.at 20μg/mL, 250nm), incorrect fonts format for „Tosyl phenylalanyl chloromethyl ketone (TPCK).”

Author Response

We appreciate you for the constructive critiques and suggestions that we have used to revise the manuscript into an improved submission. The manuscript has been revised based on your requests and comments . A response to each point is given as attached. 

Reviewer 2 Report

Article
Extraction and Isolation of Bioactive Coumarinolignans from the Roots of Waltheria indica

A brief summary
Authors should think about what they actually want to write an article about. The text presented here consists of several loosely related parts bearing a title that does not relate to the content presented. Also, the keywords only partly relate to the content of the article. The order and titles of the chapters are unusual.

Broad comments
    1. The title suggests that an optimised process for extracting compounds with known biological activity from herbal raw material will be proposed, investigated and discussed. Meanwhile, the complete and meticulous process of extracting specific (and at the same time new) compounds is presented in the methodology of the work and the Authors do not discuss it. How did the Authors know what to look for and what procedures to follow to extract just such compounds? Was there any prior screening? What, after all, is the order and nomenclature of the chemical compounds studied? Why the new compounds are numbered 5 and 7 and the previously known number 6?
    2. The main part of the work is the identification of compounds isolated from the raw material, which is not mentioned at all in the title.
    3. The order and titles of the chapters are unusual. Is the chapter entitled 'Experimental section' the 'Materials and Methods' chapter? Why make it so complicated? 
        3.1) The chapter 'General Experimental Procedures' contains a list of various pieces of equipment but without assigning them to specific research tasks or operating parameters used. It is also not clear whether it is complete, e.g. how was the chromatographic separation described in line 90 carried out?
        3.2) If this is a methodology chapter, why does it contain information such as that placed in lines 325-358? These seem to be more like results.
        3.3) Why is the chapter 'Computational Methods' placed at the end when the results related to it are discussed at the beginning of the paper?
    4. The data presented in Figures 7 and 8 have not been statistically tested, so the conclusions presented in the 22, 24, 248, 256 and 413 lines cannot be considered true.
    5. Since the structure of the investigated compounds has been explained in such detail in this work, perhaps the mechanism of their action can already be presented? (line 265)

Specific comments
Table 2. The columns were supposed to represent compounds 5 and 7 and are described as 6 and 8.

Author Response

(The authors gave the same response as above.)

Reviewer 3 Report

The manuscript ID: molecules-1655366 deals with the identification of new coumarinolignans from roots of an Hawaiian endemic plant. The newly identified compounds were fully characterized by mean of NMR. HRMS, IR and ECD spectroscopy. Finally, considering the known biological activities reported in traditional medicine for plant extracts, the compounds were evaluated in vivo for antioxidant and anti-inflammatory activities. 

To my opinion, the manuscript needs some minor revisions concerning language style and the parts highlighted need to be rewritten and rearranged as suggested in the attached copy. 

Author Response

(The authors gave the same response as above.)

Round 2

Reviewer 2 Report

The amendments made by the authors have added value to the article. However, it is still not possible to verify the results of the statistical analysis in Figures 8 and 9 (NIR, homogeneous groups).

Author Response

Thanks so much. The manuscript has been revised based on your comment.
